# Redox Responsive Copolyoxalate Smart Polymers for Inflammation and Other Aging-Associated Diseases

**DOI:** 10.3390/ijms22115607

**Published:** 2021-05-25

**Authors:** Berwin Singh Swami Vetha, Angela Guma Adam, Azeez Aileru

**Affiliations:** 1Department of Foundational Sciences and Research, School of Dental Medicine, East Carolina University, 1851 MacGregor Downs Road, MS 701, Greenville, NC 27834, USA; swamivethab20@ecu.edu; 2Physio/Biochem/New Product Development Division, Cocoa Research Center Institute of Ghana, P.O. Box 8, New Tafo-Akim 0233, Eastern Region, Ghana; angelaadamworld@gmail.com

**Keywords:** copolyoxalate, smart polymer, reactive oxygen species, polymeric drug delivery, nanodrug delivery, aging-associated diseases

## Abstract

Polyoxalate (POx) and copolyoxalate (CPOx) smart polymers are topics of interest the field of inflammation. This is due to their drug delivery ability and their potential to target reactive oxygen species (ROS) and to accommodate small molecules such as curcumin, vanilline, and p-Hydroxybenzyl alcohol. Their biocompatibility, ultra-size tunable characteristics and bioimaging features are remarkable. In this review we discuss the genesis and concept of oxylate smart polymer-based particles and a few innovative systemic delivery methods that is designed to counteract the inflammation and other aging-associated diseases (AADs). First, we introduce the ROS and its role in human physiology. Second, we discuss the polymers and methods of incorporating small molecule in oxalate backbone and its drug delivery application. Finally, we revealed some novel proof of concepts which were proven effective in disease models and discussed the challenges of oxylate polymers.

## 1. Introduction

Aging-associated disease (AAD) is an umbrella term that describes simple and complex pathological conditions which are only observed in people of advanced age in both male and female affecting one or more organs [1]. Oxygen-containing molecules with an uneven number of electrons, the reactive oxygen species (ROS), are a highly reactive by-product of the cell metabolism. With age the cell machinery loses its ability to compensate the free radicals of the ROS and antioxidant balance leading to oxidative stress [2]. Impaired reduction and oxidation (redox) balance causes oxidation of proteins, DNA damage, lipid peroxidation, and signal transduction pathway interference by ROS. To mitigate the effect of these harmful ROS, the approach was to use small molecules to scavenge and reduce free-radical formation. Redox responsive oxalate smart polymers are remarkable because of their ability to incorporate small molecules, target ROS, their biocompatibility and ultra-size tunable characteristics, as well as their ability to generate nano bubble. The objective of this review is to elucidate oxalate polymers as a choice of drug delivery system by incorporating small molecules such as curcumin, vanilline, and p-Hydroxybenzyl alcohol in the polymer backbone. 

### 1.1. Aging Associated Disease

Understanding the mechanisms involved in aging is an arduous task. A shift in balance between free radicals and oxidants within the cell depends upon its genetic makeup. Genes play a decisive role in aging and determining the redox balance. While genetic association with AAD is beyond the scope of this work, the production of free radicals and ROS are AAD-linked. It is difficult to differentiate normal aging and AAD, but free radicals and ROS are a key culprit in majority of the cases [3].

### 1.2. Free Radical Species

Reactive species includes free radicals, reactive oxygen intermediates (hydrogen peroxide (H_2_O_2_), superoxide (O_2_^•–^), singlet oxygen (^1^O_2_), peroxyl (RO^•^_2_), alloxyl (RO^•^), the hydroxyl radical (^•^OH), reactive nitrogen intermediates—nitric oxide (NO^•^), and nitrogen dioxide (NO^•^_2_)) [4]. All the above-mentioned species are metabolic byproducts formed by the metabolism of oxygen in mitochondria (Figure 1). In addition to metabolism, xenobiotic catabolism and cosmic radiations can induce the generation of these reactive species. Under normal physiological condition, intracellular reactive species plays a vital role in cell signaling [5]. 

H_2_O_2_ is more stable than other reactive oxygen and nitrogen species. It also has the ability to percolate into the surrounding extracellular spaces through aquaporin diffusion [6]. This makes it more dangerous than other reactive species [7]. Based on the “Seed and Soil” hypothesis proposed by Dr. Stephen Paget, cancer cells (the seeds) need appropriate microenvironment (the soil) to develop, spread, and metastasize systemically [8], an abnormal intracellular production of H_2_O_2_ maintains cancer phenotype, aggravates inflammation, induces liver injury, myocardial infarction, and Alzheimer’s disease [9,10,11,12]. 

### 1.3. Bio Active Small Molecules

Small molecules with bioactive principles have become an object of mystery to biomedical scientists. The Egyptians, who were advanced medical practitioners of their time, were aware of the medicinal properties of complementary medicinal formulations prepared from herbal remedies [13]. Newman et al. (Figure 2) suggested that the utilization of natural products and/or their novel structures, to discover and develop the final drug entity, is still alive and well [14,15,16,17,18]. However, nearly 40% of the newly identified natural products-based drugs widely approved for clinical translation are poorly soluble in water, hence hindering the active biodistribution of the drugs [19]. Delivering these small molecules into the human body systemically and targeting them to be activated at the site of inflammation is a meaningful approach. 

### 1.4. Systemic Drug Delivery and Prodrugs 

Systems of systemic drug delivery by enteral or parenteral route to achieve maximum therapeutic outcomes with minimal invasive procedures have gained significant momentum and attention [20]. Administered drugs can be absorbed through the gastrointestinal tract or introduced intravenously. Any conventional drug administered through the enteral or parenteral is in an active form (before biotransformation) [21]. Therefore, an actively circulating drug molecule exhibits lower efficacy, and in cases of a toxic active form, it could cause a systemic adverse effect, that is, affecting organs and organ systems. For example, 4-hydroxycyclophosphamide (HCY) is an anticancer agent (metabolite) which is a potent alkylator and potentially more toxic than its prodrug form. On the contrary the prodrug is like a Trojan horse, that stays in circulation and remains inactive. It only biotransforms inside the cells that contain the activation signal or the organ that requires the drug. Small molecules in prodrug form improve absorption, distribution, metabolism, and excretion (ADME) and at the same time decrease the toxic effect to the system. For example, cyclophosphamide (CP) is the prodrug of its active metabolite, HCY, cytochrome P450 activates CY to HCY. This reduces the adverse effect of the native drug and minimizes end-organ toxicity [22]. In recent years a new class of prodrug family known as polymeric prodrug is emerging as a powerful delivery means capable of sustained therapeutic and theragnosis. 

### 1.5. Smart Polymers

Smart polymers are novel polymers that are designed to undergo reversible or irreversible changes chemically or physically when subjected to single or multiple physiochemical or biochemical stimuli. For example, Poly (acrylic acid) is commonly used to prepare drug delivery systems. A change in pH causes ionization of the pendant acidic group resulting in rapid change in net charge and alteration in the molecular structure [23]. 

## 2. History of Polyoxalate and Copolyoxalate

Polyoxalate was first described by Ueda et al. in the early 1970s [24]. Due to its bio compatibility, it found its application in absorbable coating for sutures [25]. They contain the –O(CO)_2_O– functional group and have shown a relatively high degradability under aqueous conditions into oxalic acid and corresponding diols. About a decade ago, Murthy’s group [26] described hydrogen peroxide’s capability of sensing of polyoxalate through chemiluminescence; by early 2010 and his team reported polyoxalate polymer-based biocompatible and biodegradable particle formulation for drug delivery application [27].

### 2.1. Polyoxalate Based Drug Delivery for Inflammation

Lee’s group pioneered and designed oxalate-based polymeric prodrug formulations (Figure 3) [26,28]. CPOx smart polymers incorporated with small molecule (containing diol functional groups) could generate polymers with prodrug capability. These polymers are fabricated into micro/nano formulations and used in various biomedical applications [4]. They effectively target macrophages in inflammation and other AAD. 

Based on the biomedical application, oxalate-based polymers can be classified into first, second, and third generations of CPOx formulation (Table 1). The generation-based classification is solely based on biomedical application and there may be some overlap.

#### 2.1.1. Polyoxalate Formulations

These are polymers designed with redox responsiveness polyoxalate group. The molecular weight of the polymers was approximately in the range of 11,000 Da. These particles were early proof of concept and were designed to target macrophages in the cell.

#### 2.1.2. First Generation Copolyoxalate Formulation (Gen1 CPOx) 

These are polymeric particles designed with a redox responsive-polyoxalate group. In addition, a small molecule was covalently linked to the polymer backbone. The precursor is a co-polyoxalate polymer, hence this is considered the first generation of co-polyoxalate nano formulation. These particles were sophisticated in comparison to their polyoxalate predecessors. Since the formulations were in their infancy no systemic delivery applications were carried out. 

#### 2.1.3. Second Generation Copolyoxalate Formulation (Gen2 CPOx)

Second generation of copolyoxalate nanoformulation are polymeric particles derived from the previous generation’s redox responsiveness concept. However, the biomedical applications of these polymers were extended to in vivo models. 

#### 2.1.4. Third Generation Copolyoxalate Nanoformulation (Gen3 CPOx)

The third generation of copolyoxalate nanoformulation are polymeric particles designed to be theragnostic in nature. These particles delivered therapeutic small molecules in an in vivo animal models and were simultaneously imaged after the drug was delivered.

## 3. Bio Compatibility of Polyoxalate and Copolyoxalates 

Drug delivery systems comprised of biodegradable polymeric nanocarriers are based on FDA approved poly d,l-lactic-co-glycolic acid (PLGA), and are used in the continual release of loaded small molecules in the system. However, the slow hydrolysis kinetics of PLGA results in an accumulation of acidic degradation product and induces inflammation. The limitation of PLGA encouraged Lee’s group to develop POx polymer. In the presence of an aqueous solution, POx degrades to form non-toxic byproducts, 1,4-cyclohexanedimethanol (CHDM), oxalic acid, and CO2 (Figure 4). POx was designed to degrade rapidly in the presence of high concentration of H_2_O_2_ [27,29]. CHDM and oxalic acid can be readily metabolized by the human body without causing any adverse effects. An in-vitro study of molecular toxicity of CHDM in human cells showed that it did not bind to estrogen and androgen receptors. In rat models it was demonstrated that 300 mg/kg-day was observed to cause no adverse teratogenic or lethal effects. Interestingly, it was found that the fetus and placenta were more resistant to CHDM. They were able to withstand a dose of 1000 mg/kg-day [30]. Bioassay of both in-vivo and in-vitro studies indicate that CHDM exhibits low toxicity, no mutagenicity, and weak teratogenicity [30]. Oxalic acid, on the other hand, is a byproduct of tricarboxylic acid cycle and glyoxylate and methylcitrate cycles. Healthcare providers recommend 40–50 mg of oxalate-rich food for dietary consumption [31]. The oxalate concentration in the nanoformulation (15–20 µg) is significantly less than the recommended dietary oxalate intake and can be safely metabolized. Oxalic acid present in the polyoxalate nanoparticles can be absorbed in the kidneys and excreted as urine or it can form insoluble calcium oxalate and be eliminated in the feces [32].

Biocompatibilities of POx and CPOx nanoparticles were studied with PLGA nanoparticle formulation. Cell viability of POx nanoparticles and PLGA nanoparticles were carried out in RAW 264.7 cells Pox; in comparison to PLGA, nanoparticles POx nanoparticles exhibited a significantly higher cell viability demonstrating POx polymers superiority over PLGA (Appendix A) [27]. CPOx and PLGA nanoparticles were injected in thigh muscle, and the cytomorphological changes followed by the exposure of the particle to the interstitial muscle were studied (Appendix A). In the PLGA treated group there were notable changes in the volume composition of the interstitial space due to inflammation. CPOx did not exhibit inflammation or fibrosis which validates its superiority over PLGA [33].

In vivo animal experiments in mice were performed by injecting 3 mg/kg per day of CPOx and PLGA nano formulations. In the PLGA-treated group, there were inductions of NADPH oxidase 2 which signifies induction of intracellular ROS. There was no alteration of intracellular NADPH oxidase 2 activity (Appendix A) [34]. CPOx nanoparticles treatment systemically showed that there were no adverse reactions induced by the nano formulations in the histology of TUNNEL (Appendix A) [35]. Biomedical application of the Pox and various CPOx are summarized in Appendix A. 

### 3.1. Gen1 Copolyoxalate

The concept behind the CPOx polymers and their corresponding nano formulation were from the polyaspirin polymer, a study conducted by Uhrich and co-workers [36]. POx polymers do not contain any small molecule in their polymer backbone, whereas all generations of CPOx are polymeric prodrugs. These polymeric prodrug systems were uniquely engineered to incorporate the small molecules into the backbone of the CPOx. Erdmann et al. established that incorporation of small molecules in the polymer backbone, via an appropriate synthesis methodology, can yield up to 100% of the weight of small molecules loaded in the polymer assembly.

p-Hydroxybenzyl alcohol is extracted from *Gastrodia elata* and it is an aromatic diol compound incorporated in oxalate backbone. p-Hydroxybenzyl alcohol-incorporated copolyoxalate (HPOX) was the first CPOx synthesized. HPOX nanoformulation was evaluated in RAW 264.7 macrophage cells with lipopolysaccharide (LPS)-induced inflammation. Protein expression studies confirmed that p-Hydroxybenzyl alcohol from the HPOX nanoparticles exhibited anti-inflammatory and antioxidant activity [37,38].

Poly(amino oxalate) (PAOX) is another example of Gen1 CPOx which is a cytosolic delivery vehicle that covalently incorporates piperazinediethanol in its backbone. PAOX is hydrophilic and has a cationic charge due to the tertiary amines present in the piperazinediethanol. Its application in a disease model for specific cytosolic protein target is yet to be explored [39].

### 3.2. Gen3 Copolyoxalate

Third generation CPOx nanoformulation contains small molecules such as curcumin, vanilline, p-Hydroxybenzylalcohol in the backbone with high-level biomedical application, i.e., theragnosis. They deliver the small molecule at the site of inflammation, which is simultaneously tracked and imaged in real-time. Biomedical application of these nanoformulation of Gen3 CPOx are discussed below. 

#### 3.2.1. Gen3 Copolyoxalate to Treat Liver Inflammation

Aging increases the vulnerability of the liver to acute liver injury and fibrotic response. The severity and poor prognosis of various liver diseases deteriorate with age and thereby increases ROS load in the liver. Delivering a small molecule that could scavenge the excessive ROS could protect the liver from the harmful ROS and restore redox homeostasis [40].

The primary constituent of vanilla pods (*Vanilla planifolia*) is vanilla (4-hydroxy-3-methoxybenzaldehyde) which is peculiar for its culinary and medicinal properties. Rapid degradation of the vanillin through the oral rout or through the intravenous route diminishes the therapeutic outcome of vanillin. Poly(vanillin oxalate) (PVO) a polymeric prodrug of vanillin, incorporates the small molecules covalently in its backbone and release its drug cargo in pathologically escalated levels of H_2_O_2_ and hydrolytic degradation (Figure 5). When PVO particles encounter cells undergoing oxidative stress, they readily degrade and release vanillin resulting in localized antioxidant and anti-inflammatory activity. PVO nanoparticles in vitro and in vivo distribution through systemic delivery proved excellent antioxidant activities by scavenging H_2_O_2_, thereby, inhibiting the generation of ROS and the expression of pro-inflammatory cytokines [41].

Curcuminoid is a bioactive principle from the plant tuber of *Curcuma longa*. It is known for its therapeutic capability such as anti-aging, antioxidant, anti-inflammatory, and anti-cancer ability [28]. The therapeutic potential of curcumin is appreciated due to the poor water solubility property of the small molecule and low systemic bioavailability as well as molecular instability. Hydrogen peroxide-activatable poly(oxalate-co-curcumin) (POC) polymeric prodrug of curcumin incorporates the curcumin small molecule in the backbone of POC (Figure 6). POC was synthesized by a one-step polymerization reaction of curcumin (20 mol%), CHDM (80 mol%), and oxalylchloride (100 mol%). The idea of preparing polyoxalate polymer was to covalently incorporate curcumin in the polymer backbone through peroxalate ester linkages. The average molecular weight of the synthesized POC was ~8500. POC particles were prepared from the POC polymer by employing single emulsion technique. In in vitro condition, RAW 264.7 cells stimulated with H_2_O_2_ and treated with POC particles internalized particles through endosomes and suppressed intracellular ROS. Systemic delivery of the POC particles in acetaminophen (APAP)-induced liver failure remarkably reduced the level of alanine aminotransferase (ALT) (Appendix A); the therapeutic effect was superior to free small molecules of curcumin (Appendix A). POC particle treatment did not elicit any adverse tissue reaction. ALT and terminal deoxynucleotidyl transferase dUTP nick-end labeling (TUNEL) assay results of the POC particle treated group showed no ALT induction or damage to the liver tissue (Appendix A). In vivo echogenic performance of POC nanoparticles performed in a study showed aggregation of the particle formulation in the liver which confirms the delivery of the small molecule at the site of inflammation (Appendix A) [42].

#### 3.2.2. Gen3 CPOx to Treat Muscle and Tendon Injuries

Skeletal muscle is one of the largest organs in the body, irrespective of sex. It is subjected to wear and tear with age. Often physiological changes depend upon the physical activity of the individual. Skeletal muscle is one of the largest consumers of oxygen in the body: approximately 20% per minute, and hence superoxide anions are produce at several sites and released to the muscle cells’ cytosol [43,44]. With advance in age skeletal muscles are subjected to poor antioxidant defense resulting in muscle deterioration. Externally ingested antioxidants could inhibit the rate of oxidation, establish redox balance, and improve the quality of waning muscles. However, long-term administration of mitochondria-targeted antioxidants was not proven to be effective in age-related skeletal muscle dysfunction [45]. A polymeric prodrug of CPOx could help in unlocking the redox potential of therapeutic small molecules. 

Poly (vanillyl alcohol-co-oxalate) PVAX-based nano formulation was evaluated in contusion injury of triceps surae muscles and Achilles tendons. The study showed that PVAX nano formulation exhibit anti-inflammatory activity by reducing macrophage accumulation at the site of injury. Protein expression of caspase-3, Bcl-2, and Bax confirms the anti-apoptotic potential for the PVAX particle formulation. The integration of diagnosis and therapeutic nature of PVAX, using ultrasound imaging of musculoskeletal injuries was tested and proved to be an added advantage to age-related skeletal muscle dysfunction [46].

#### 3.2.3. Gen3 CPOx to Treat Airway Inflammatory Diseases

Many people live in an air polluted environment where people with high sensitivity are susceptible to foreign bodies. The inflammatory cells defense is sensitized by the foreign bodies which leads to mucoid secretion and shortness of breath. Delivering small molecules that could exhibit antioxidant and anti-inflammatory could mitigate the ill effects of ROS in the lung. Biocompatible CPOx-containing PVAX polymeric particles could help in easing the inflammation. An in vivo study performed with porous PVAX microparticles loaded with allergy medicine dexamethasone (DEX), followed by intratracheal injection of particles, showed a rapid drug release in porous PVAX microparticles loaded with DEX. DEX combined with vanillyl alcohol was shown to enhance therapeutic outcome [47]. Since the lungs have a large surface area for absorption, this CPOx micro/nano carriers could be explored further for sustained delivery of the therapeutic proteins or peptides from lung to systemic circulation.

#### 3.2.4. Gen3 CPOx to Treat Cardiac Diseases

Ischemic tissue after blood perfusion is subjected to ROS insults that can overwhelm redox balance, damage the surrounding tissues, and induce apoptosis. Biodegradable hydroxybenzyl alcohol containing polyoxalate copolymer (HPOX) with multifunctional capability was used by Lee et al. in their study to evaluate HPOX’s therapeutic efficacy. HPOX’s particles showed localized antioxidant and anti-apoptotic activity at the site of inflammation. In vivo hind-limb reperfusion injury treated with HPOX’s loaded with 4-amino-1,8-napthalimide showed reduced poly ADP ribose polymerase-1 and caspase-3 protein expression which corresponds to the antioxidant and anti-inflammatory properties of HPOX [48]. In another study, antioxidant vanillin containing poly(vanillin oxalate) (PVO) was utilized to treat ischemia/reperfusion (I/R) injury. The PVO nanoparticles were intravenously administrated, systemically localized in the liver, and were proven to suppress inflammation and apoptosis and inhibited the liver damage progression. As with any other Gen3 CPOx, PVO was imaged simultaneously by ultrasound which was helpful in confirming the localization and delivery of the therapeutic load [49].

Lee et al. and Bae et al. investigated the vanillyl alcohol incorporated PVAX in myocardial I/R (ischemia/reperfusion) injury. The first objective of the study was to provide an insight on the biocompatibility of the potential for PVAX formulation [50]. The later study described the therapeutic effect of the systemically delivered formulation capability to deliver the therapeutic load at the site of myocardial I/R injury and the chemiluminescence imaging revealed the localization of the particles at the I/R injury site (Figure 7). In addition, PVAX significantly reduced the level of NADPH oxidase (NOX) 2 and 4 expression, which favors the reduction in ROS generation after I/R. [34]. Both studies demonstrated the theragnostic potential of the PVAX as a therapeutic agent for myocardial I/R injury. HPOX, PVAX, and PVO particle formulations hold the translational capability to be used in targeting ischemia, cardiovascular, and neurovascular drug delivery simultaneously and deliver therapeutic small molecule to the inflammatory site [51].

#### 3.2.5. Gen3 CPOx to Treat Cancer

According to the World Health Organization’s Global Health Observatory, in 2018 alone, 9.6 million people are estimated to have died from cancer, and 30–50% of cancer incidences could have been prevented [52,53]. As previously mentioned, in free radicals and reactive species subdivision, cancer cells are triggered to be in a hyper drive state. When the cells are in hyper drive state they proliferate and consume more nutrients and oxygen, which leads to escalated ROS generation. However, the cancer cells’ defense mechanisms counteract the ROS induced apoptosis. A localized induction of ROS that pushes the mutated cancer cells beyond its redox balancing capability is a potential domain where Gen3 CPOx comes into action. 

Photodynamic therapy is a well-known treatment modality that is helpful in treating cancer. However, due to its penetration depth limitation its application cannot be fully appreciated in cancer therapy. Our study “Novel chemi-dynamic nanoparticles (CDNP)” showed that there is a light-free photodynamic therapeutic system for cancer treatment [54]. The mechanism of action of CDNP is that it utilizes the highly stable reactive species H_2_O_2_ as a trigger. The naturally occurring small molecules were pre-treated to sensitize the cancer cells, followed by photosensitizer molecule encapsulated CDNP. H_2_O_2_ present in the cancer cells permeates and enters the CDNP core containing HPOX and protoporphyrin IX (PPIX). This rapidly gives rise to high energy 1,2-dioxethanedione intermediate. Energy released from the decomposition of these unstable 1,2-dioxethanedione results in chemi-dynamic therapy (CDT), i.e., PPIX activation, ^1^O_2_ generation, and results in programmed cell death of the cancer cell (Figure 8). The study performed established the enhanced tumor therapy by combining free small molecule and polyoxalate based CDNP to enhance CDT induced targeted cancer therapy. This was the first report on light-free photodynamic therapeutic system based on polyoxalate polymers which was proven to induce apoptotic cell death in cancer cells [55]. Moreover, around the same time Andrey et al. reported similar mechanism. They elaborated on the ROS generated by the CDNP, singlet oxygen generation in cell-free system and in cell culture system [56].

Liu’s research group pushed the application of the CDNP much further, in their work. They used Bis[2,4,5-trichloro-6-(pentyloxycarbonyl)phenyl] oxalate (CPPO) encapsulated with a specially designed photosenzitiser TPE-BT-DC (Figure 9). In vivo, β-phenylethyl isothiocyanate infusion was used to increase the H_2_O_2_ concentration in the breast-tumor and then the CPPO containing particle was administered systemically. Enhanced permeability and retention of the nano formulation increase the possibility of the nanoparticles to be activated at the site of the cancer cells. Particle accumulation served as an image-guided tumor therapy aided in fluorescence and chemiluminescence bioimaging of the tumor [57]. In comparison to other inflammatory diseases CPOx-based novel particle formulations for light free tumor therapy, which is relatively in its infancies, further evaluation in other cancer models could help to discern an effective clinical translation of CDT.

## 4. Limitation of the POx and CPOx

FDA approved PLGA has been used in the field of biomaterial since the early 1960s. POx and CPOx are relatively new polymeric biomaterials. Despite its advantages POx and CPOx material synthesis involves a complicated process and various hazardous solvents. To be commercially viable, methodology of the synthesis must be modified and simplified. Translation of these smart polymer are reduced due to limited investment in research and development to improve the methods of POx and CPOx synthesis. It is hoped that this review would shed a positive light over POx and CPOx and gain traction in biomedical sciences.

## 5. Conclusions

Polymeric prodrug has gained a considerable amount of interest as a naturally occurring phenolic compound incorporated in CPOx, an emerging avenue of prodrug mediated drug delivery. Within the past decade, POx and CPOx based novel micro and nano formulations were intensively investigated in inflammatory diseases ranging from wound to light free tumor therapy. In the future we anticipate (a) CDT’s application in animal models and human trials; (b) POx and CPOx nano formulation that could cross the blood–brain barrier and treat ailments that are as a result of redox imbalance in brain; (c) multi targeted nano formulation with more narrow targeted action; and (d) POx and CPOx particle formulations from bench to bedside.

## Figures and Tables

**Figure 1 ijms-22-05607-f001:**
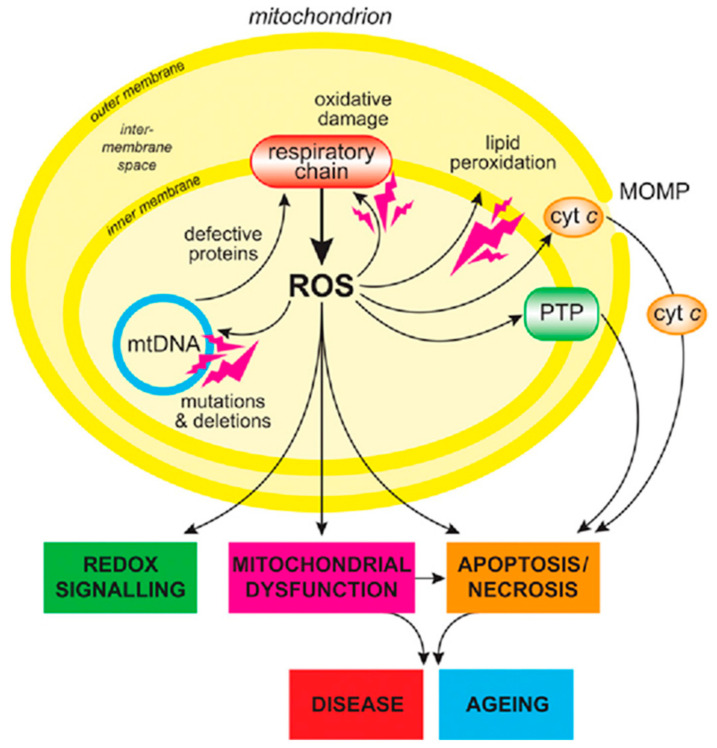
Genesis of ROS in mitochondria and its fate [5]. Cells apoptotic cycle is activated when mitochondrial DNA (mtDNA) undergoes damage due to the increased intra-mitochondrial ROS surge. The sequence of events triggered after oxidative damage are: (1) Cytochrome C permeates out of mitochondria by the outer membrane permeabilization (MOMP), (2) intra mitochondria ROS surge fuel the production of mitochondrial permeability transition pore (PTP). This event increases the susceptibility of the mitochondrial permeability to the small molecules leading to cell apoptosis and subsequent AAD.

**Figure 2 ijms-22-05607-f002:**
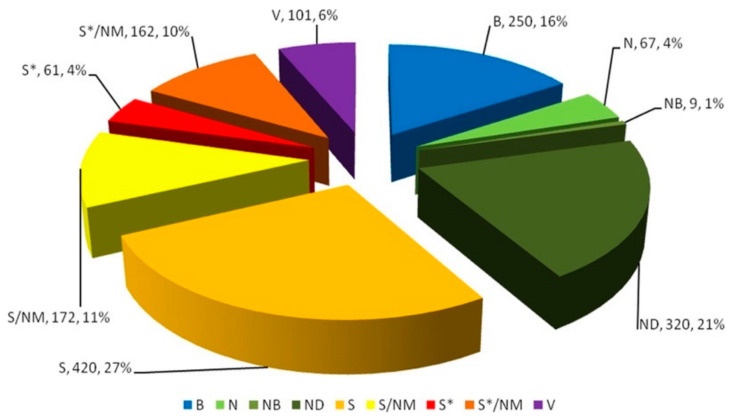
All approved drugs between 1981–2014 that were based on small-molecule; *n* = 1211. B: Biological macromolecule, N: Unaltered natural product, NB: Botanical drug (defined mixture), ND: Natural product derivative, S: Synthetic drug, S*: Synthetic drug (NP pharmacophore), V: Vaccine, /NM: Mimic of natural product [14] Reprinted with permission.

**Figure 3 ijms-22-05607-f003:**
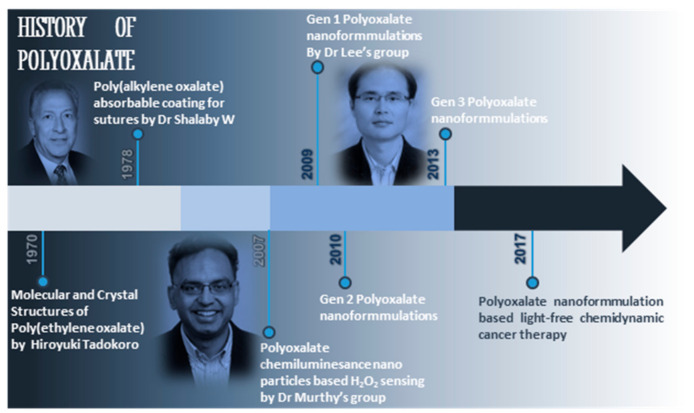
History of polyoxalate and co-polyoxalate.

**Figure 4 ijms-22-05607-f004:**
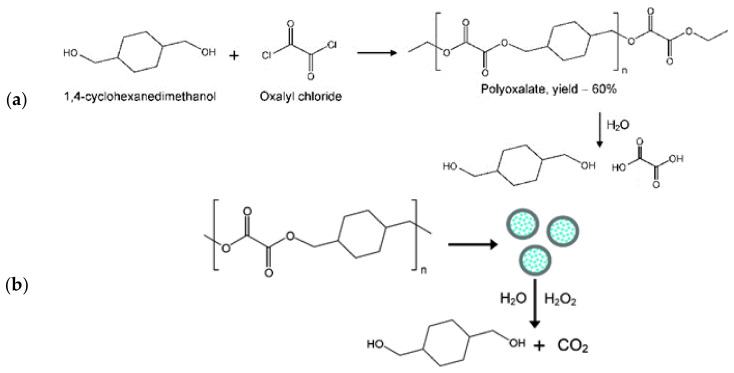
(**a**) Polyoxalate polymer synthesis and degradation, (**b**) nanoformulation preparation and their biodegradation [27] Reprinted with permission.

**Figure 5 ijms-22-05607-f005:**
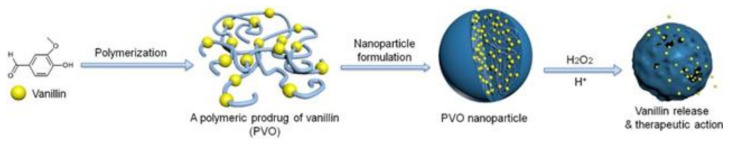
PVO Nanoparticles preparation and degradation [41] Reprinted with permission.

**Figure 6 ijms-22-05607-f006:**
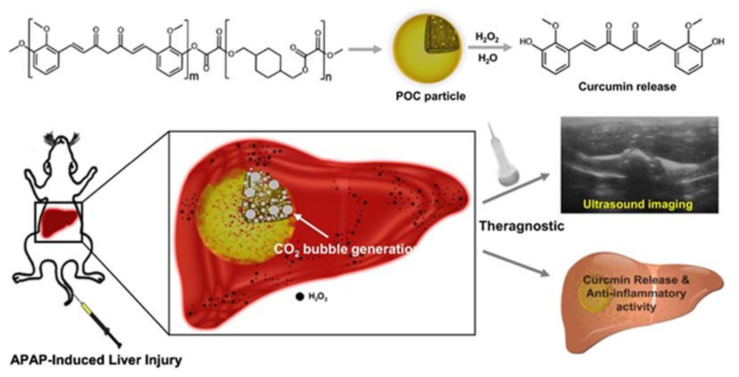
Theragnostic POC particles for acute liver failure [42] Reprinted with permission.

**Figure 7 ijms-22-05607-f007:**
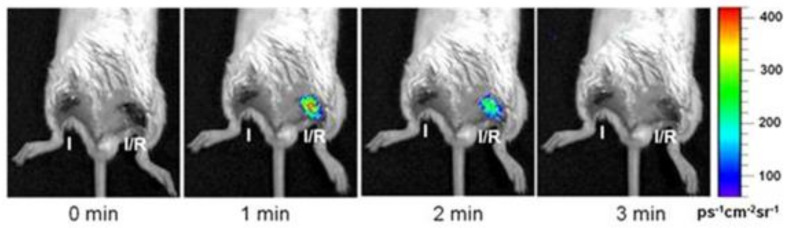
Bioluminescence imaging of PVAX loaded with rubrene’s in vivo after I/R in mouse hind-limbs [50] Reprinted with permission.

**Figure 8 ijms-22-05607-f008:**
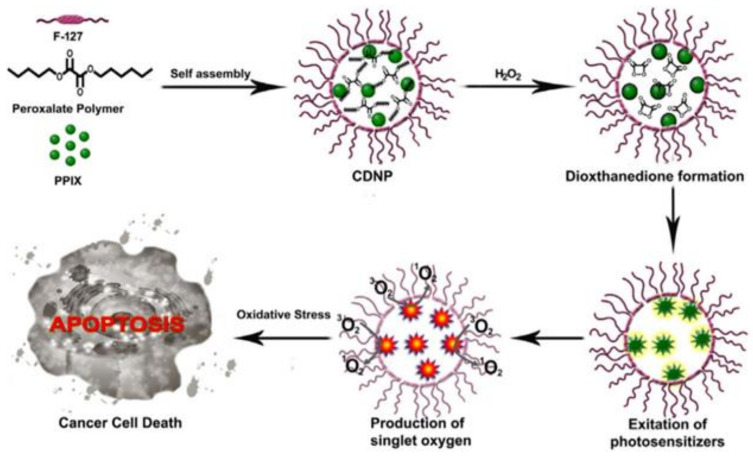
Light-free chemidynamic therapy using CDNP nanoreactors [55] Reprinted with permission.

**Figure 9 ijms-22-05607-f009:**
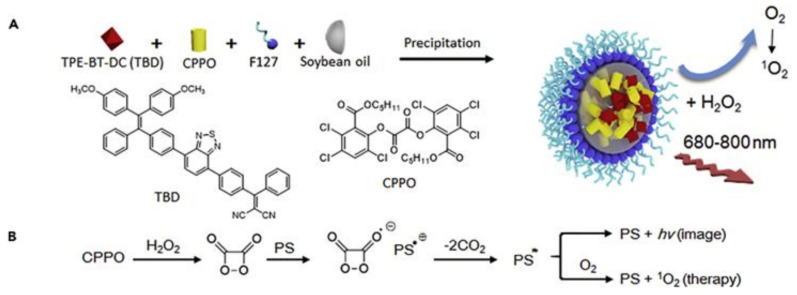
The preparation of chemiluminescence nanoparticles (**A**) Chemiluminescence nanoparticle preparation (**B**) principle for chemiluminescence and ^1^O_2_ generation [57] Reprinted with permission.

**Table 1 ijms-22-05607-t001:** Polyoxalate, oxalate, and copolyoxalate-based formulations classification.

	Generation 0	Generation 1	Generation 2	Generation 3
Covalent prodrug incorporation	×	✓	✓	✓
Redox response	✓	✓	✓	✓
Drug loading capability	✓	✓	✓	✓
Micelle self-assembly incorporation	✓	✓	✓	✓
in vitro evaluations	✓	✓	✓	✓
in vivo evaluations	×	×	✓	✓
Systemic drug delivery	×	×	✓	✓
Theragnostic applications	×	×	×	✓
			✓—present	×—absent

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
