# Peer review of "Redox Responsive Copolyoxalate Smart Polymers for Inflammation and Other Aging-Associated Diseases"

_ijms, 2021, doi:10.3390/ijms22115607_

Round 1

Reviewer 1 Report

Major concerns:

  1. Please add a brief description of the legend of Figure 1, including the diagram about the represented figure and abbreviations which were shown in the figure.
  2. Figure 2 is not clear. Please re-upload it.
  3. It is suggested to add a small graph about the prospective application of CPOx in disease treatment, as well as, the potential difficulties in development.
  4. What is the CPx (line 344)? It didn’t appear in the previous paragraph.

Minor concerns:

  1. Please explain the meaning of dual responsiveness (Table 1).
  2. Please give the full name of APAP (line 230) and PPIX (line 318).
  3. It is suggested to make a simple table or figure to summarize the CPOx-conjugated nature compounds and their application, as well as, the benefits from the conjugation with CPOx.
  4. The subtitle of the 3rd paragraph (line 167) is suggested to be re-subtitled with CPOx.

Author Response

Response to Reviewer 1 Comments

Major concerns:

Point 1: Please add a brief description of the legend of Figure 1, including the diagram about the represented figure and abbreviations which were shown in the figure.

Response: Abbreviation’s with additional technical description are now added to Figure 1.

Point 2: Figure 2 is not clear. Please re-upload it.

Response: An image with better clarity is presented.

Point 3: It is suggested to add a small graph about the prospective application of CPOx in disease treatment, as well as, the potential difficulties in development.

Response: Applications of the oxalate polymers are described separately in the supplement Table1 and the potential limitation of developing POx and CPOx are discussed in section 4.

Point 4: What is the CPx (line 344)? It didn’t appear in the previous paragraph.

Response: Sentence is rewritten by substituting with the correct terminology POx.

Minor concerns:

Point 1: Please explain the meaning of dual responsiveness (Table 1).

Response: Third Gen Polyoxalates includes some dual responsive nanoformulation was a “misnomer” and it is removed from the paragraph. 

Point 2: Please give the full name of APAP (line 230) and PPIX (line 318).

Response: Abbreviated terms APAP and PPIX have been defined in the manuscript.  

Point 3: It is suggested to make a simple table or figure to summarize the CPOx-conjugated nature compounds and their application, as well as, the benefits from the conjugation with CPOx.

Response: Applications of the oxalate polymers are described separately in the supplement Table 1.

Point 4: The subtitle of the 3rd paragraph (line 167) is suggested to be re-subtitled with CPOx.

Response: Section 3 was modified as “Biocompatibility of polyoxalate and copolyoxalates” for better clarity.

Reviewer 2 Report

This is an interesting review on the use of polyoxylate -biopolymers. This manuscript could benefit from re-organization and extensive re-writing to make it more attractive to the general readership. The relevance of the use of these polymers in animal models of human diseases compared to traditional biopolymers need to be presented . Toxicity -other side effects of these polymers need discussion. Perspectives need to be discussed.

Author Response

Response to Reviewer Comments

Point 1: Authors need to cite references to H2O2 percolating via aqua-Orin?

Response: Reference added.

Point 2: Sentence on line 82 needs to be rewritten.

Response: Based on the reviewer’s suggestion for a better clarity sentence rewritten.

Point 3: Ref #25 refers to Murthi’s group in the text but cites a paper by Kim et.al. This needs to be fixed.

Response: Reference rearranged.

Point 4: Suggest authors consider preparing a Table summarizing various biomedical applications of polyoxylate biopolymers with appropriate references.

Response: Applications of the oxalate polymers were described in the supplement Table1.

Point 5: Sentence on line 173 needs rewriting.

Response: Based on the reviewer’s suggestion for a better clarity sentence rewritten.

Point 6: On line 231 a better discussion on the synthesis of curcumin polyoxylate Biopolymer is needed. It’s head to head comparison with curcumin eluded to in Ref# 35 needs to be presented.

Response: Details about curcumin polyoxalate synthesis were added and additional information regarding the head-to-head comparison of the curcumin was added in the supplement file.

Point 7: Sentence 238-241 needs are -write. As well sentences 251-253.

Response: Based on the reviewer’s suggestion for a better clarity sentence rewritten.

Point 8: Provide detailed mechanistic explanation is required how and why Gen3cpOx is useful in treating diseases.

Response: For a better clarity sentences were modified in the manuscript. Gen3 CPOx and its application in liver inflammation, muscle and tendon injuries and cancer were described along with the schematic representation in figures 5 – 9. 

Point 9: Re write sentence 330-333

Response: Based on the reviewer’s suggestion for a better clarity sentence rewritten.

Point 10: What are the side effects and limitations of the use of polyoxylate biopolymers?

Response: Limitations of POx and CPOx are discussed in section 4 of the manuscript.

Point 11: Toxicity studies need to be discussed.

Response: The toxicity of the polyoxalate and copolyoxalates polymers and degradation products are included in section 3 of the manuscript.

Round 2

Reviewer 2 Report

I have reviewed the revised article.  The authors have done a good job with addressing the issues raised previously. The only suggestion I have for the authors is they discuss the limitations of the use of polyoxylate biopolymers. As oxylate rapidly binds to calcium it forms crystals known as kidney stones. Excess oxalic acid in blood can cause gout and in intestine dietary issues -collectively contributing to inflammation. Aside from this comment ,the paper is now acceptable for publication. Thanks for this opportunity.

Author Response

Point 1: I have reviewed the revised article.  The authors have done a good job with addressing the issues raised previously. The only suggestion I have for the authors is they discuss the limitations of the use of polyoxylate biopolymers. As oxylate rapidly binds to calcium it forms crystals known as kidney stones. Excess oxalic acid in blood can cause gout and in intestine dietary issues -collectively contributing to inflammation. Aside from this comment, the paper is now acceptable for publication. Thanks for this opportunity.

Response:

The concern raised by the review is valid and we thank the reviewer for pointing out this trepidation. However, the reason we do not concur with the comments regarding the oxalate toxicity is because of the following reasons:

(i) When we tested various oxalate formulations (mentioned in S4), the health of the kidney after oxalate nanoparticles were not affected i.e. particle treatment did not induce apoptosis of renal tissue. Healthcare providers recommend 40-50 mg of oxalate-rich food for dietary consumption as indicated in this finding [1]. Consequently, the oxalate concentration (15-20 µg) in the nanoformulation is significantly less than the recommended dietary oxalate intake and can be safely metabolized.

(ii) The oxalate concentration in the final nanoparticle is much less than 20 mg. The oxalate content in the polymers range from a minimal 19.7 to a maximum of 27.5 mmol. This range is attributable to a change in reactivity of the guest small molecules i.e. curcumin, vanillyl alcohol, or hydroxy benzyl alcohol, incorporated along with the polyoxalate in the polymer backbone. For example, if we prepare copolyoxalate polymers incorporated with 27 mmol of oxalate and if it is assumed to have a theoretical yield of 100%, the oxalate present in the polymer would be 2.4 grams. It should be noted that in the real-world scenario, oxalate in the polymer backbone will be lower than the above-mentioned hypothetical value. If we use 200 mg polymer (27 mmol oxalate incorporated) to prepare particles the oxalate content in the particles would be in the range of 10-15 mg. Adhering to the above analogy 100-200 µg of particles (ideally use in the in vivo studies) would contain 7.5-15 µg of oxalate in the formulation which is much lower than recommended dietary oxalate intake. 

(iii) Particles used in the majority of the in vivo models were in a range of 100-200 µg. Since the oxalate concentration is lower than the toxic concentration (>50mg), there were no adverse renal tissue responses or accumulations in the reported studies. 

(iv) At present we neither test the oxalate nanoformulation in an animal model with renal dysfunction nor acquire data that suggest nephrotoxicity. Including oxalate nanoformulation-induced nephrolithiasis will be speculative. Thus, we do not have any basis to discuss oxalate nanoformulation-induced nephrolithiasis.

  1. Mitchell, T., Kumar, P., Reddy, T., Wood, K. D., Knight, J., Assimos, D. G., & Holmes, R. P. Dietary oxalate and kidney stone formation. Am J Physiol Renal Physiol2019, 316(3): F409-F413.